# Infant Motor Milestones and Childhood Overweight: Trends over Two Decades in A Large Twin Cohort

**DOI:** 10.3390/ijerph17072366

**Published:** 2020-03-31

**Authors:** Silvia I. Brouwer, Ronald P. Stolk, Meike Bartels, Toos C.E.M. van Beijsterveld, Dorret I. Boomsma, Eva Corpeleijn

**Affiliations:** 1Institute of Sportstudies, Hanze University of Applied Sciences, Zernikeplein 17, 9747 AS Groningen, The Netherlands; 2Department of Epidemilogy, University Medical Center Groningen, PO Box 30.001, 9700 RB Groningen, The Netherlands; r.p.stolk@rug.nl (R.P.S.); e.corpeleijn@umcg.nl (E.C.); 3Department of Biological Psychology, Vrije Universiteit Amsterdam, Van der Boechorststraat 7, 1081 BT Amsterdam, The Netherlands; m.bartels@vu.nl (M.B.); t.van.beijsterveldt@vu.nl (T.C.E.M.v.B.); di.boomsma@vu.nl (D.I.B.)

**Keywords:** motor milestones, infant, childhood, overweight, BMI, secular trends

## Abstract

Background: Poor motor skill competence may influence energy balance with childhood overweight as a result. Our aim was to investigate whether the age of motor milestone achievement has changed over the past decades and whether this change may contribute to the increasing trend observed in childhood overweight. Methods: Motor skill competence was assessed in children from the Young Netherlands Twin Register born between 1987 and 2007. Follow-up ranged from 4 up to 10 years. Weight and height were assessed at birth, 6 months, 14 months, and 2, 4, 7, and 10 years. Results: Babies born in later cohorts achieved their motor milestones ‘crawling’, ‘standing’, and ‘walking unassisted’ later compared to babies born in earlier cohorts (N = 18,514, p < 0.001). The prevalence of overweight at age 10 was higher in later cohorts (p = 0.033). The increase in overweight at age 10 was not explained by achieving motor milestones at a later age and this persisted after adjusting for gestational age, sex, and socioeconomic status. Conclusion: Comparing children born in 1987 to those born in 2007, we conclude that children nowadays achieve their motor milestones at a later age. This does not however, explain the increasing trend in childhood overweight.

## 1. Introduction

Over the past decades the prevalence of overweight and obesity has increased in children [1]. Childhood overweight is of clinical interest because it tracks into adulthood [2,3] and it is associated with metabolic risk factors for cardiovascular disease [4].

According to the developmental approach of Stodden [5], the potential role of motor skill competence may promote physical activity and weight status. Therefore, motor skill competence may be one factor that explains the changes in energy balance over time, with childhood overweight as a result [6]. There are signs that motor skill competence is declining [7,8,9], but literature is sparse. Several factors may play a role. For example, to reduce the chance of sudden infant death syndrome, parents are recommended since 1987 to let their baby sleep on their back or side, instead of on their tummy. However, tummy time aids in developing neck, shoulder, and core strength, which increases the development for other motor milestones [10]. Furthermore, the arrival of equipment like maxi-cosies, babywalkers, and screens like television and i-pads may play a role [11,12]. 

Several studies describe the relationship between motor skills and overweight. In infants aged below 24 months, overweight is more often reported in those who reach their motor milestones at an older age [13,14]. Although these studies provide evidence for an association between motor skill competence and overweight, cause and consequence are uncertain, because the association can be bidirectional. On the one hand it is possible that lower scores on motor skill competence may cause overweight by not having the opportunities to be more physically active and thereby spend more energy. On the other hand, overweight may cause lower scores on motor skill competence since overweight may hinder movement. Prospective studies on the association between motor skill competence and overweight in children show that lower scores on motor skill tests are associated with increased BMI [15,16] suggesting causality. For infants, the situation is somewhat different. Infants may have more difficulty in starting to walk when they have a higher body weight-for-length, because it takes more strength. On the other hand, if they have developed faster and stronger, a higher body weight may indicate better growth and muscle mass, and this may favor achievement of motor milestones. Indeed, there is some evidence for associations in both directions. Although babies with lower birth weight (BW) seem to reach their motor milestones later compared to those with higher BW [17,18], two prospective studies with infants younger than 18 months found that overweight predicts lower scores on motor skill tests [14,19]. In addition, lower scores on motor skill tests predict a higher sum of skinfolds in later childhood (age 3 years) [20] but no relation was found between time of achieving motor milestones and BMI at the age of 7 [18]. The results are thus inconclusive, which warrants further investigation. One way to address this question is to look at trends in achievement of motor milestones and overweight over time, using a cohort effect approach. 

The aim of our study was 4-fold: (1) to examine changes over the past twenty years in motor milestone achievement and growth in body weight (weight-for-length, BMI, and overweight) by comparing children from birth years 1987 to 2007; (2) to study the association between early growth (weight-for-length) and the age that infants achieve motor milestones; (3) to study the association between motor milestones and childhood BMI at ages 2, 4, 7, and 10 years; and (4) to examine whether changes in overweight could be explained by later motor milestone achievement. Data from twins participating in the Young Netherlands Twin Register (YNTR) were used to address these aims, because the YNTR is a unique cohort with a continuous influx of new-born twins since 1987.

## 2. Materials and Methods 

*Population:* Data from twins participating in the Young Netherlands Twin Register (YNTR) were analyzed in this study. The YNTR was established in 1987 as a population-based volunteer register. It recruits families with twins a few months after birth and registers around 40% (ranging from 38% to 53% over the years) of all multiple births in the Netherlands [21]. Prospective data have been collected for twins since 1987 through age-specific surveys in which parents provide data on development and health. Because data from children from the same family are dependent, one twin from each pair was selected randomly. Inclusion criteria were available data on motor milestones and BW and gestational age (GA). Exclusion criteria were preterm birth (<32 weeks), very low BW (<1500 grams), and disability or illness that interferes with daily functioning. All data used in the analyses were collected under protocols that have been approved by the appropriate ethics committees, and this stutabdy was performed in accordance with the ethical standards from the 1964 Declaration of Helsinki and its later amendments.

*Birth data:* After registration of new-born twins, mothers received an initial survey with items on pregnancy and birth. Mothers provided information on GA, BW, and length. To support keeping track of motor milestones, a table was attached to the first survey for parents to enter data when motor milestones were reached (Table 1). 

*Age 0–2 years:* A second survey on growth and motor milestone achievement was sent at age 2. Parents were asked to record when motor milestones were reached: rolling over from back to belly, sitting without support, crawling on hands and knees, standing without support, walking without support, and speaking for the first time. Obtaining data on motor milestone achievement by mail using retrospective surveys on children turning 2 years of age has been shown to be a reliable method [22]. In a validation study, data for motor milestones conducted via monthly telephone interviews from 217 twin pairs were compared to retrospective mail surveys from 463 twin pairs born in the same year. Motor milestone achievement reported by monthly telephone interviews showed no differences compared to retrospective data from mail surveys. For data on growth, parents of twins were asked to provide the report with growth data measured by nurses from the Youth Health Services from birth to age 2 (13 visits until the age of 2 years: 4 and 8 weeks, 3, 4, 6, 7.5, 9, 11, 14, and 18 months, 2 and 3 years, and 3 years and 9 months). For this study, growth data of BW 6 months (prior to the mean age of achieving the first of motor milestone), 14 months (just before mean age of achieving the last motor milestone), and 2 years are used. 

*Age 4–10 years:* For the growth data on 4, 7, and 10 years, length and weight were reported in age-appropriate questionnaires [21]. Growth data for 4 years were obtained in the same way as growth data between birth and age 2. Growth data for 7 and 10 years were measured by parents. 

*Socioeconomic status*: Parents were asked to report the family socioeconomic status (SES), measured as highest parental occupation at ages 4, 7, and 10 of their twins. Family SES was scored in five different categories that approximately translate to: (1) ‘unskilled labor’, (2) ‘job for which lower vocational education is required’, (3) ‘job at medium level’, (4) ‘job at college level’, and (5) ‘job at university level’ [23]. 

*Data calculations:* For weight and length until age 14 months, weight-for-length (WfL) standard deviation scores (SDS) were calculated using the Growth Analyzer 3 software package, with the Dutch reference growth charts for the general population from 1997 [24]. A score below 0 means having a lower WfL compared to the reference group and a score above 0 means having a higher WfL compared to the reference group. BMI at age 2, 4, 7, and 10 years was calculated as weight (kg) divided by length (m) squared. Standard deviation scores were calculated using the same software package. Linear interpolation was used to reduce the amount of missing data as follows: missing data for BMI SDS at age 2 years were interpolated based on the BMI SDS at age 14 or 18 months and at 3 years; missing data for BMI SDS at age 4 years were interpolated based on the BMI SDS at age 2 or 3 years and 5 or 7 years; and missing data for BMI SDS at age 7 years were interpolated based on the BMI SDS at age 4 or 5 years and at 10 years. For age 2, 4, and 7 years, 1467 (10.7%), 2410 (33.5%), and 1027 (14.3%) data points, respectively, were interpolated this way. Most of the data missing at 10 years is because children born in the latest cohorts did not reach the age of 7 or 10 years during the study yet. They have a shorter follow-up; therefore, the number of children with growth data at age 7 and 10 is lower compared to the number of children at younger ages. Children were classified as overweight or obese using Cole’s extended international body mass index cut-offs classification [25].

*Data analyses:* SPSS 20.0 was used for the analyses. Infants were subdivided into seven birth cohorts (1987–1989, 1990–1992, 1993–1995, 1996–1998, 1999–2001, 2002–2004, and 2005–2007). To examine differences for motor milestone achievement, GA, BW, birth length, WfL SDS, and BMI SDS across birth cohorts, an ANOVA was used. The chi-square was used to test the differences in prevalence of overweight (yes, no) age 4, 7, and 10. To study the association between early growth and motor milestones, multiple linear regression analysis was used with the different motor milestones as the dependent and BW and WfL age 6 or 14 months as the independent. The analyses were adjusted for GA, exact age during measurement of growth, sex, socioeconomic status (SES), and cohort. To study the association between motor milestones and childhood BMI SDS at ages 2, 4, 7, and 10 years linear regression analyses was used with BMI SDS at ages 2, 4, 7, or 10 years as the dependent and the different motor milestones as the independent while adjusting for GA, exact age during measurement of growth, sex, socioeconomic status (SES), and cohort. Regression coefficients (β values) and 95% confidence intervals (CI) are reported. To examine whether an increase in overweight prevalence could be explained by later motor milestone achievement, logistic regression was used with overweight (yes/no) at age 2, 4, 7, or 10 years as the dependent variable. In our first model, cohort was used as the independent. In our second model, we adjusted for the covariates GA, exact age during measurement of WfL or BMI, sex, and socioeconomic status (SES). In our third model, “moment of walking” was added. In the fourth model, the interaction between “moment of walking” and cohort was added. Within the results and discussion we mainly use the term infants and/or children to refer to our twin population.

## 3. Results

In total, data for 20,367 children from the Netherlands Twin Register were included. Infants without data on GA or BW, with preterm birth (<32 weeks GA), or with very low BW (<1500 g ) were excluded (n = 1625, 8%). Infants with a disability or illness that interferes with daily functioning were also excluded (n = 22, 0.1%). A total of 18,514 individuals remained for the final analyses. 

In Table 2, the characteristics of the children are shown. Mean GA was 37 ± 2.0 weeks. At 6 and 14 months, the infants in this study had a negative mean WfL SDS, indicating that mean WfL was lower than for the reference group (general population). Mean BMI-for-age at 2, 4, 7, and 10 years were all negative (between −0.1 and −0.4 SD) and therefore lower compared to the reference group. An ANOVA showed that there were no differences in WfL SDS age 6 and 14 months and BMI SDS at age 2, 4, 7, or 10 years between children with 0 till 6 missing values on growth. However, there was a linear association between missing data and GA and BW in the way that children with more missing data (0 till 6) on growth had shorter GA (p < 0.001) and weighed less compared to children with less missing data on growth (p < 0.001).

### 3.1. Cohort Effects 

GA decreased between 1987 and 2007 from 37.4 weeks in the first cohort to 36.8 weeks in the last cohort (p < 0.001), as reported before [26]. A decrease in GA from 1987 to 2007 was not accompanied with a decrease in BW, indicating a general trend towards a higher BW for a given GA. Although in this study preterm birth (<32 weeks) were excluded, changes in BW depended on GA: up to 32 weeks, birthweight decreased, and after 32 weeks, birthweight increased [26]. Although there was a significant cohort effect for the age at which ‘rolling over’ was achieved (Figure 1A; p < 0.001), there was no specific linear time trend for an increase or decrease, but infants in cohort 4 and 5 reached the moment of rolling over earlier compared to infants form the other cohorts. Infants born in the last cohort achieved crawling, standing, and walking later than infants born in the earlier cohorts (Figure 1A + B; p < 0.001). This linear trend was most pronounced for the motor milestones ‘standing without support’ and ‘walking without support’. Infants born in 1987–1989 reached these milestones almost one month earlier than infants born in 2004–2007 (from 11.7 ± 2.4 to 12.6 ± 2.6 months for standing without support and from 14.5 ± 2.1 to 15.3 ± 2.4 months for walking without support). The ‘first word spoken’ also differed across cohorts (Figure 1B; p < 0.001) but showed an opposite linear trend. Infants born in the latest cohorts spoke their first word earlier than those born in the earlier cohorts, with the exception of the first cohort. WfL SDS at age 6 and 14 months and BMI SDS age 2 (Figure 1C), 4, and 7 years (Figure 1D) showed significant differences across cohorts (respectively: p < 0.001; p < 0.001: p = 0.010; p < 0.001; p = 0.043), but there was no obvious linear trend in any direction with exception for BMI SDS age 4 years. At the age of 4 years, BMI SDS was lowest in cohort 2 and increased from cohort 2 to cohort 6. However, there were no significant differences between the first and last cohort for BMI SDS age 4 years. With regard to the prevalence of overweight (yes/no), an increase in the prevalence of overweight was observed at the age of 10 years (*χ*^2^ (df = 4, N = 5325) = 10.457; p = 0.033) but not at age 4 or 7 years (Figure 1E).

### 3.2. Association between BW and Early Weight-for-Length and Motor Milestone Achievement

First, we investigated whether body weight at very young age is a determinant for achieving milestones later, from the perspective that with a higher weight, movements may take more effort and thereby may be related to a later onset of weight carrying movements. Table 3 shows that a lower BW is associated with achieving all motor milestones later (rolling: β = −0.235 [95%CI: −0.311; −0.160] p < 0.01; sitting: β = −0.130 [−0.208; −0.052] p<0.01; crawling: β = −0.125 [−0.228; −0.023] p < 0.01; and standing: β = −0.157 [−0.266; −0.048] p < 0.01), except for ‘walking without support”. Infants with a relatively low weight for their length SDS at age 6 or 14 months reached most but not all motor milestones later compared to infants with higher WfL SDS at age 6 (sitting: β = −0.148 [−0.179; −0.117] p < 0.01; standing: β = −0.099 [−0.143; −0.055] p < 0.01; and walking: β = −0.111 [−0.151; −0.071] p < 0.01;) or 14 months (sitting: β = −0.149 [−0.185; −0.112] p < 0.01; crawling: β = −0.063 [−0.111; −0.016] p < 0.01; standing: β = −0.150 [−0.201; −0.099] p < 0.01; and walking: β = −0.165 [−0.212; −0.119] p < 0.01). An exception is made for ‘rolling over’. For ‘rolling over’, there was an association in the other direction at age 6 months, in the way that infants with higher WfL SDS reached their moment of rolling over later (β = 0.055 [0.025; 0.085] p < 0.01). All these analyses were adjusted for GA, gender, exact age of assessment, SES, and cohort.

### 3.3. Association between Motor Milestone Achievement and Childhood Overweight at Age 2, 4, 7, and 10 Years

Next, the association was studied from the other perspective, that it is possible that later achievement of motor milestones may be associated with overweight at later ages, based on the assumption that lower scores on motor skill competence are related to lower levels of PA and therefore less energy expenditure. In general, later achievement of motor milestones is related to lower values for BMI SDS. This was significant for ‘sitting without support’ for all ages (age 2: β = −0.023 [−0.033; −0.012] p < 0.01; age 4: β = −0.019 [−0.033; −0.005] p < 0.01; age 7: β= −0.032 [−0.046; −0.018] p < 0.01; and age 10: β = −0.039 [−0.056; −0.021] p < 0.01), and for the all other milestones with BMI SDS at 7 (crawling: β = −0.012 [−0.023; −0.001] p<0.05; standing: β = −0.023 [−0.033; −0.012] p < 0.01; and walking: β = −0.024 [−0.035; −0.013] p < 0.01) and 10 years of age (crawling: β = −0.016 [−0.030; −0.003] p < 0.05; standing: β = −0.029 [−0.042; −0.017] p < 0.01; and walking: β = −0.031 [−0.045; −0.018] p < 0.01). Again, the exception is for ‘rolling over’. At age 2 years and not age 4, 7, and 10 years, infants who achieve the moment of rolling later have higher BMI SDS (β = 0.014 [0.003; 0.025] p < 0.05) (Table 4). The analyses were repeated on the original data. No differences were found between the results of the interpolated data and the original data with the exception of the association between the motor milestone “walking” and BMI age 4 years. In the interpolated data only a trend was visible (β = −0.009 [−0.020; −0.001] p < 0.08), and in the original data this association was significant (β = −0.014 [−0.026; −0.003] p < 0.01).

To determine whether changes in motor milestone achievement over two decades may partly explain the growing prevalence of overweight at age 10, the logistic linear regression showed that this increase in overweight over cohorts could not be explained by the later achievement of motor milestones when adjusting for GA, exact age during measurement of WfL or BMI, sex, and socioeconomic status (SES) (β = −0.043; p = 0.068).

## 4. Discussion

Comparing children born in 1987 to those born in 2007, we conclude that children nowadays achieve their motor milestones at a later age. Over the same era, overweight at age 10 tended to increase with time. The later age of motor milestone achievement did not however, explain the increasing trend in childhood overweight. Furthermore, WfL and BMI-for-age SDS at different ages did not increase in twins born between 1997–2007. In contrast to what was expected, infants with lower BW and WfL at ages 6 and 14 months reached their motor milestones slightly but significantly later compared to heavier infants, and infants who reached their motor milestones later have slightly but significantly lower BMI.

The later achievement of motor milestones found in this twin population was for the motor milestones crawling, standing, and walking and not for rolling and sitting. The findings in the first cohorts, 1987 to 2001, of the YNTR population were published earlier [9], and our present study shows a consistent linear trend that continues until 2007. The later achievement of motor milestones was most pronounced for standing and walking without support. Infants born in 2007 achieved standing and walking on average one month later than infants born in 1987. Although the stability over time in age of motor milestone achievement we found in twins for rolling and sitting is similar to the stability found over twenty years in singletons by Darah et al. (2013) [27], the later achievement of other motor milestones we found in twins was not found in another study in singletons by Darah et al. (2014) [28]. Differences in how and when motor skill competence is measured could explain the differences in outcomes. The authors used the Alberta Infant Motor Scale (AIMS) [29]. The AIMS does not report the actual age a motor milestone is achieved. Instead, the infant is observed at a specific moment to determine whether he or she is able to perform various motor actions. It is therefore more difficult to detect a change in age of achievement. Furthermore, the AIMS does not observe motor skill competence after standing with support and does not evaluate crawling. Our study found delays in motor skill competence specifically in crawling, standing, and walking without support. It is possible that delays are only latent in later motor milestones that were measured by our study but not by Darah et al. [28]. With regard to the use of data of twins, the comparison on motor milestone achievement between twins and singletons must be discussed. We expect that the outcomes can be generalized towards singletons because within the normal window of achievement of motor milestones there are no major differences between twins and singletons, as we have shown before [9]. Differences in the age of milestone achievement, are mainly explained by the shorter GA of twins. Therefore, when studying motor milestones in our twin population, we adjusted for GA. Furthermore, babies born pre-term (before 32 weeks of GA) or with very low BW (<1500 grams) are considered at risk for motor development deficits [30] and are not included in our studies. Although no other studies were found investigating a cohort trend in motor skills competence in infants, there is support for a decrease in motor skills competence in older populations. Preschoolers as well as school-aged children born in later cohorts score lower on motor skill competence than children born in earlier cohorts [7,8]. Because motor skill competence tracks across time [31] the delay in motor milestone achievement we found in twins is in line with decreased motor skill competence scores in older populations [7,8].

It seems clear that children develop their motor skill later nowadays than 20 years ago. The explanation may be found in the guideline to avoid prone position during sleep, to reduce the risk of sudden infant death. Indeed infants with more time in prone position during sleep or being awake reach their motor milestones earlier [32,33]. It may also be related to changes in our transportation habits, as it seems that restriction to move freely in infants could be the reason for later achievement of motor milestones [34].

The question that rises is how this affects the activity level and health of the children. Our study shows that babies with lower BW achieve rolling, sitting, crawling, and standing later than those born with higher BW. The associations were weak but consistent for all motor milestones except walking. The consistency in the associations strengthens the conclusion that babies with lower BW achieve their motor milestones later compared to heavier babies. Comparable results for an association between lower BW and later age of achieving motor milestones were found in the WHO child growth standards, which collected data on motor milestones of 816 infants from six different countries [35], the Northern Finland birth cohort [17], and the Danish National Birth Cohort [18]. It can be concluded that babies with lower BW reach their milestones later compared to babies with higher BW after adjusting for GA. When investigating associations between weight status after birth but before or during all motor milestones are reached, we found that lower WfL at age 6 and 14 months was associated with later achievement of motor milestone. These results are in line with the results found in the Danish cohort [18] but is in contrast with the study of Slining et al. (2010) [19] which found that thicker skinfolds were associated with later achievement of motor milestones. Prospective associations between early growth and later motor skill competence seem limited or absent.

Because several studies found that, based on BMI, overweight children score lower on motor skill competence test compared to normal weight children [19,36], but evidence from prospective studies do not show consistent evidence for an association between motor skill competence and overweight [15,16,18,20], we expected that infants who reach their motor milestones later would have higher BMI during childhood. In contrast, we found that infants who reach their motor milestones later have slightly lower BMI-for-age SDS at ages 2, 4, 7, and 10 years. Because other prospective studies are performed in population with different ages, we compared our study specific with studies measuring motor skill competence during infancy. This association between later achievement of motor milestones and lower BMI at later age was also found in The Northern Finland birth cohort study [17]. In this study, BMI at age 14 years was slightly lower in those who achieved their moment of walking later. In addition, in a large Danish cohort later achievement of motor milestones was associated with slightly lower BMI at age 7 years [18]. In another population, late achievers had smaller WfL at age 3 years, which is in line with our study [20]. However, the authors showed that late achievers had higher adiposity when measured by skin-fold thickness. The authors suggest that BMI is arguably not sensitive enough to detect associations between anthropometrics and motor skill competence because skinfolds thickness has better associations with body fat compared to BMI [37,38]. Although we found significant associations between motor milestones and subsequent BMI, the association was small and clinically not relevant. Together with previous studies [17,18,39], we suggest no association between motor milestones and subsequent BMI during childhood.

Regarding the worldwide increase in overweight in children [1], it is noteworthy that our study only showed a minimal increase in overweight at age 10 years. The decrease in GA from 1987 to 2007 was not accompanied with a decrease in BW, indicating a general trend towards higher BW for a given GA. This is in line with the increases of BW of singletons in the same period (1989–2006) [40]. This study shows that BMI in twins at ages 2, 4, 7, and 10 seems relatively stable over the last twenty years. The prevalence of overweight and obesity in this study at ages 4, 7, and 10 is lower than for reference populations [41]. It is known that there are differences in length/height, weight, and BMI between twins and singletons, which decline with increasing age but do not disappear completely [42,43,44]. This could partly explain the lower than expected prevalence of overweight during childhood.

Strength and limitations: The strength of our study is the large population with data available for motor milestone achievement in different cohorts (born between 1987–2007) and a follow-up up to 10 years for > 5000 children. The limitation is that not all children weight status was available at all time points. Children with more missing data on weight status had shorter GA and lower BW, which could have caused a bias in the way that these children were more at risk for divergent growth patterns. In addition, only twins participated in our study. Although there are differences in growth between singletons and twins [42,43,44], we do not expect that these differences could affect our conclusions since the associations between motor milestones and growth were clinically not relevant and comparable to studies in singletons. It is known that babies born with lower BW are at higher risk for developing overweight [45]. However, this is normally for babies born with (very) low BW, which we excluded from our study. Furthermore, including maternal characteristics like age, BMI, and way of conceiving, might be of interest but are currently outside the scope of the present paper.

## 5. Conclusions

In conclusion, infants born in later cohorts achieve their motor milestones later than infants born in earlier cohorts. The achievement of motor milestones is minimally associated with BMI suggesting that motor milestone achievement and BMI are largely independent and that the delay in motor milestone achievement does not explain an increase in childhood overweight.

## Figures and Tables

**Figure 1 ijerph-17-02366-f001:**
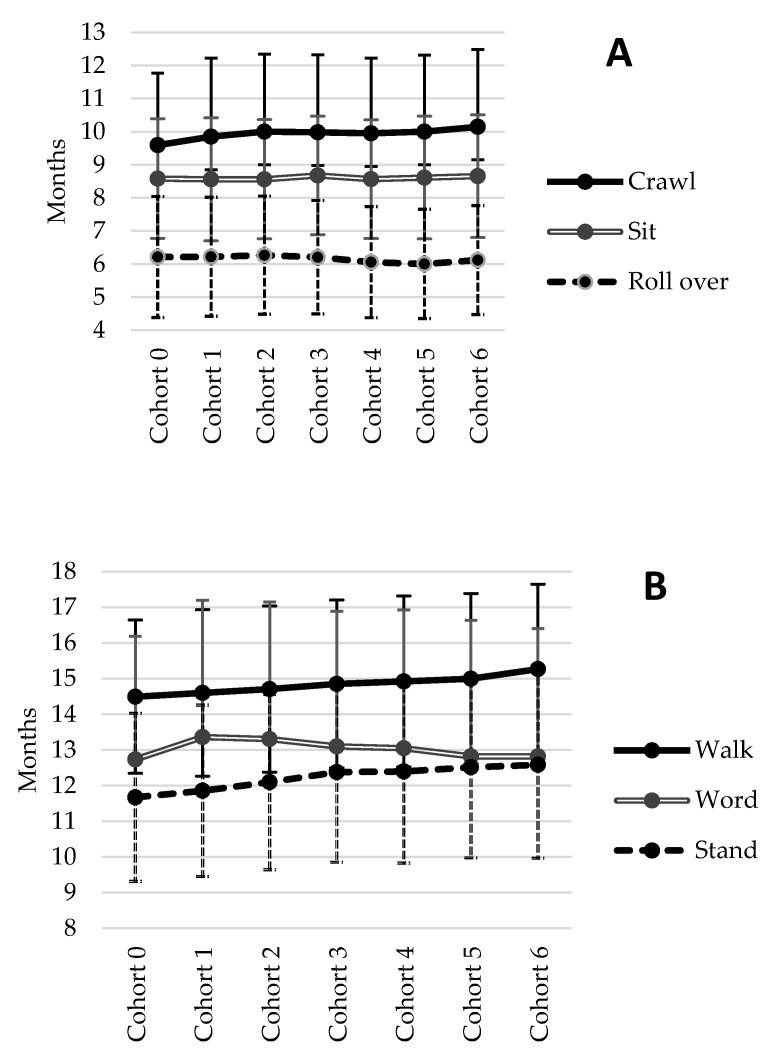
Cohort-specific means and standard deviations for motor milestones (**A**) rolling, sitting, crawling, (**B**) standing, walking, and speaking the first word, (**C**) WfL SDS age 6, 14 months and BMI SDS at 2 years, (**D**) BMI SDS at 4, 7, and 10 years, and (**E**) prevalence overweight at age 4, 7, and 10 years in twins from the Netherlands Twin Registry (NTR).

**Table 1 ijerph-17-02366-t001:** Data collection of twins from the Young Netherlands Twin Register (YNTR).

	Survey
*Age 0–6 months*	Pregnancy and birth Table for keeping track of motor milestones
*Age 2 years*	Growth and motor milestone achievement
*Age 4–10 years*	Growth and socioeconomic status

**Table 2 ijerph-17-02366-t002:** Baseline descriptive statistics of twins from the Young Netherlands Twin Register (YNTR).

	Birth	6 months	14 months	2 years	4 years	7 years	10 years
**Infants**							
Total (N)	18,514	15,619	12,998	12,261	6793	6152	5325
Girls, (N)	9282	7857	6568	6186	3461	3176	2750
Gestational age (weeks)	37.0 ± 2.0	-	-	-	-	-	-
Age (years)		0.5 ± 0.0	1.2 ± 0.1	2.1 ± 0.2	3.9 ± 0.2	7.3 ± 0.3	10.0 ± 0.5
**Anthropometrics**							
Weight (kg)	2.6 ± 0.5	7.1 ± 0.8	10.1 ± 1.1	12.6 ± 1.5	16.6 ± 2.1	25.1 ± 4.0	33.8 ± 6.1
Length (cm)		65.5 ± 2.7	77.5 ± 3.1	88.3 ± 3.8	104.3 ± 4.4	127.6 ± 6.0	143.2 ± 7.1
WfL (SDS)		−0.05 ± 0.98	−0.09 ± 0.93	-	-	-	-
BMI (kg/m^2^)		-	-	16.2 ± 1.3	15.2 ± 1.3	15.4 ± 1. 8	16.4 ± 2.2
BMI-for-age (SDS)		-	-	−0.10 ± 1.02	−0.32 ± 1.03	−0.40 ± 1.06	−0.31 ± 1.10
Overweight/obese (%) *		-	-	-	5.4	7.7	8.2

Population totals are presented in numbers; Gestational age, age at day of measurements and anthropometrics are presented as means ± standard deviation. WfL (Weight-for-length) and BMI-for-age are presented as standard deviation scores (SDS) and overweight/obese as percentage. * Children were classified as overweight or obese using Cole’s extended international body mass index cut-offs classification [25].

**Table 3 ijerph-17-02366-t003:** Associations between birth weight and weight-for length SDS at 6 and 14 months and motor milestones.

		Rolling		Sitting		Crawling		Standing		Walking	
Age of achievement	(months)	6.2 ± 1.7		8.6 ± 1.8		9.9 ± 2.3		12.2 ± 2.5		14.8 ± 2.3	
Birth weight (grams)	birth	−0.235	[−0.311; −0.160] **	−0.130	[−0.208; −0.052] **	−0.125	[−0.228; −0.023] **	−0.157	[−0.266; −0.048] **	−0.085	[−0.184; 0.015]
Weight-for-length SDS	6 months	0.055	[0.025; 0.085] **	−0.148	[−0.179; −0.117] **	−0.030	[−0.071; 0.011]	−0.099	[−0.143; −0.055] **	−0.111	[−0.151; −0.071] **
	14 months	0.016	[−0.007; 0.063]	−0.149	[−0.185; −0.112] **	−0.063	[−0.111; −0.016] **	−0.150	[−0.201; −0.099] **	−0.165	[−0.212; −0.119] **

Data are presented as mean ± SD or β and 95% confidence interval and are adjusted for GA, actual age, sex, SES, and cohort; * p < 0.05, ** p < 0.01.

**Table 4 ijerph-17-02366-t004:** Associations between motor milestones and BMI SDS at 2, 4, 7, and 10 years.

	Age of Achievement (Months)	BMI SDS			
		age 2 years	age 4 years	age 7 years	age 10 years
Rolling	6.2 ± 1.7	0.014 [0.003; 0.025] *	0.012 [−0.002; 0.026]	−0.001 [−0.016; 0.014]	−0.011 [−0.029; 0.007]
Sitting	8.6 ± 1.8	−0.023 [−0.033; −0.012] **	−0.019 [−0.033; −0.005] **	−0.032 [−0.046; −0.018] **	−0.039 [−0.056; −0.021] **
Crawling	9.9 ± 2.3	0.001 [−0.007; 0.009]	0.000 [−0.010; −0.010]	−0.012 [−0.023; - 0.001] *	−0.016 [−0.030; −0.003] *
Standing	12.2 ± 2.5	−0.003 [−0.010; 0.005]	−0.013 [−0.022; −0.003] **	−0.023 [−0.033; −0.012] **	−0.029 [−0.042; −0.017] **
Walking	14.8 ± 2.3	−0.005 [−0.013; −0.003]	−0.009 [−0.020; −0.001]	−0.024 [−0.035; −0.013] **	−0.031 [−0.045; −0.018] **

Age of achieving motor milestones is presented as mean ± SD. BMI standard deviation scores (SDS) are presented as β and 95% confidence interval and are adjusted for GA, actual age, sex, SES, and cohort; * p < 0.05, ** p < 0.01.

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
