# Peer review of "Infant Motor Milestones and Childhood Overweight: Trends over Two Decades in A Large Twin Cohort"

_ijerph, 2020, doi:10.3390/ijerph17072366_

Round 1

Reviewer 1 Report

This is a comparative study where authors tried to investigate whether the age of motor milestone achievement has changed over the past decades and whether this change may contribute to the increasing trend observed in childhood overweight. They concluded that comparing children born in 1987 to those born in 2007, children nowadays achieve their motor milestones at a later age, without this to explain the increasing trend in childhood overweight.

There are several points to be revisited before consideration this paper as suitable for publication, including the lack of rationale for encompassing only twins and the accompanying not proper explanations in the limitations section; the lack of explanation of the “random” selection of one of them; the lack of association of the authors findings and parameters with confounding factors; the comparisons of evidence coming from different type and quality of studies; the existence of theories for explanations not based in high quality studies,

Author Response

Response to Reviewer 1

Dear reviewer,

Thank you very much for reviewing our paper and for the considerations to improve our paper. We will response to your comments and questions. 

There are several points to be revisited before consideration this paper as suitable for publication, including:

Point 1: the lack of rationale for encompassing only twins.

Response point 1: Unfortunately, the twin cohort we used did not include singletons. Of course, data on singletons would add extra information regarding generalizability.  However, despite this lack of singletons, we considered it worthwhile to perform this analysis. The Young Netherlands Twin Register (YNTR) is a cohort study with continuous influx of new-born twins since 1986, resulting in a unique dataset to study changes in development over time, with data from newborns starting in 1987. Although there are cohorts with singletons collecting data on motor milestones and body weight, these studies are mostly closed cohorts, meaning they include children over a period of 1 to 3 years, and then follow them up over time. We added in the introduction the text “because the YNTR is a unique cohort with a continuous influx of new-born twins since 1987” [line 71].

Point 2: Not proper explanations in the limitations section;

Response point 2: To explain the use of twin data we wrote in the discussion section the text “We do expect that the outcomes can be generalized towards singletons because within the normal window of achievement of motor milestones there are no major differences between twins and singletons[ref 8: 8. Brouwer SI, van Beijsterveldt TC, Bartels M, Hudziak JJ, Boomsma DI: Influences on achieving motor milestones: a twin-singleton study. Twin Res Hum Genet 2006, 9(3):424-430. ]. Differences in the exact age of milestone achievement, can mainly be explained by the shorter GA of twins. When studying motor milestones in our twin population, we adjusted for GA. Furthermore, babies born pre-term (before 32 weeks of GA) or with very low BW (<1500 grams) are considered at risk for motor development deficits[ref 31: Allen MC, Alexander GR: Gross motor milestones in preterm infants: correction for degree of prematurity. J Pediatr 1990, 116(6):955-959.] and are therefore not included in our studies”. Although within the normal window of achievement of motor milestones there are no major differences between twins and singletons, there are differences between singletons and twins in the exact age of the moment to walk without support. We compared the age of achievement of “walking without support”of twins with singletons from the Groningen Expert Center for Kids with Obesity (Gecko) Drenthe cohort (Brouwer SI, Stolk RP, Corpeleijn E. Later achievement of infant motor milestones is related to lower levels of physical activity during childhood: the GECKO Drenthe cohort. BMC Pediatr. 2019;19(1):388. Published 2019 Oct 28. doi:10.1186/s12887-019-1784-0). Differences between twins and singletons in the exact moment of achieving walking without support was dependent on year of birth the range of gestational age we took into account (for example 36-41 weeks or 36-38 weeks or comparison with exact the same gestational age rounded to half a week) were varying from 0.1 to 1.0 months. Singletons from the Gecko Drenthe cohort achieved walking without support with 14.1 ± 1.9 (mean gestational age 39.9 ± 1.6) while twins were able to walk with 14.8 ± 2.3 (mean gestational age 37.0 ± 2.0), varying from 14.5 ± 2.1 to 15.3 ± 2.4 when born in 1987-2000 and 2005-2007 respectively.  Therefore, when adjusting for gestational age, we assume that using motor milestones from twins is not a limitation but discussed it using the above described text in the discussion [line 259-265].

Point 3: The lack of explanation of the “random” selection of one of them

Response point 3: We realize that we didn’t gave a proper explanation for the random selection. Because we wanted independent observations we randomly selected one of each twin-pair. This means that we did not take into account characteristics of the twins like birth order, heaviest baby, gender of opposite-sex twin etc.

Point 4: The lack of association of the authors findings and parameters confounding factors;

Response point 4:

It is not clear what the exact question is. Is it a lack of an association between motor milestones and BMI/overweight, the time trend or other associations? Can the question be rephrased or explained? That would help us to formulate an answer.

Point 5: The comparisons of evidence coming from different type and quality of studies;

Response point 5: It would be ideal to compare the outcomes of our study with other high quality studies. With regard to developments in motor development over time, the number of studies is limited. Regarding the main findings of our study focusing on later achievement of motor milestones, we only found the studies of Darah et al. (Darrah J, Bartlett DJ: Infant rolling abilities--the same or different 20 years after the back to sleep campaign? Early Hum Dev 2013, 89(5):311-314.; and Darrah J, Bartlett D, Maguire TO, Avison WR, Lacaze-Masmonteil T: Have infant gross motor abilities changed in 20 years? A re-evaluation of the Alberta Infant Motor Scale normative values. Dev Med Child Neurol 2014, 56(9):877-881.) which were published in the first and second quartile journals. No other studies were found to compare with (reference 26 and 27).

Point 6: The existence of theories for explanations not based in high quality studies

Response point 6: We added in the introduction Stodden’s model (2008) who describes the role of motor development in relation to physical activity and overweight. Pathways of this model were tested in the past years and evidence for several pathways are described in the review of Robinson (2015) (Robinson LE, Stodden DF, Barnett LM, et al. Motor Competence and its Effect on Positive Developmental Trajectories of Health. Sports Med. 2015;45(9):1273–1284. doi:10.1007/s40279-015-0351-6). Based on evidence of Stodden’s model it can be suggested that the associations between infant motor skill competence and subsequent PA and overweight might get stronger with increasing age. To illustrate this, if infants reach their motor milestones later, they have less opportunities to move around, explore and develop new skills. They will have lower levels of PA and higher BMI. Indeed we found that infants who reach their motor milestones later have lower levels of subsequent PA (Brouwer SI, Stolk RP, Corpeleijn E. Later achievement of infant motor milestones is related to lower levels of physical activity during childhood: the GECKO Drenthe cohort. BMC Pediatr. 2019;19(1):388. Published 2019 Oct 28. doi:10.1186/s12887-019-1784-0).  Because of these lower levels of PA, these children have less opportunities to develop new or more specified skills (Lopes L, Santos R, Pereira B, Lopes VP: Associations between sedentary behavior and motor coordination in children. Am J Hum Biol 2012, 24(6):746-752). We added the text: “According to the developmental approach of Stodden [5], the potential role of motor skill competence may promote physical activity and weight status [line 33-34].

Reviewer 2 Report

General Comment

This paper is based on infant motor milestone comparing different time points. This is an interesting topic specially nowadays due to the advent of new technologies with greater contact with children.  

Specific Comments

-    Line 54, 55, 56, 57, 58: The phrase “Although babies with… …motor skill tests.” is confuse;

-    Line 64: “…BMI and overweight); by…” remove semicolon from the phrase;

-    Line 65: “…from birth years 1987 to 2007, 2) to…” change comma to semicolon;

-  The authors correlate motor milestones with overweight. However, overweight also involve family genetic background and, as mentioned in the introduction, overweight may lower scores on motor skill competence. On the other hand, studies suggest that babies with high birth weight compared to low birth weight may reach their motor milestone earlier. How the authors address the motor milestone taking the family genetic background in mind?

-   In the methods, authors mentioned that data had been collected for majority of twins selected in this study. But what majority means in percentage?

-    Why authors used twins for the study?

-    Did the family sign an agreement with the survey?

-    The authors didn’t mention whether the study analyzes or compares normal delivery with C-section. Are there any differences?

-    In the methods, as a suggestion, authors should include a new table organizing the survey during ages;      

-    In the conclusion, line 235, “Ove the same are…” should be “Over the…”.

Author Response

Reviewer 2

Dear reviewer,

Thank you very much for reviewing our paper and for the considerations to improve our paper. Thank you for sharing the opinion that this paper describes an interesting topic. We will response to your comments and questions below. 

Point 1: Line 54, 55, 56, 57, 58: The phrase “Although babies with… …motor skill tests.” is confuse;

Response point 1: We rephrased the sentence to: “Although babies with lower birth weight (BW) seem to reach their motor milestones later compared to those with higher BW[17, 18],  two prospective studies with infants younger than 18 months found that overweight predicts lower scores on motor skill tests [Line 55-57].

Point 2: Line 64: “…BMI and overweight); by…” remove semicolon from the phrase;

Response point 2: We removed the semicolon.

Point 3: Line 65: “…from birth years 1987 to 2007, 2) to…” change comma to semicolon;

Response point 3: We changed the comma to semicolon.

Point 4: The authors correlate motor milestones with overweight. However, overweight also involve family genetic background and, as mentioned in the introduction, overweight may lower scores on motor skill competence. On the other hand, studies suggest that babies with high birth weight compared to low birth weight may reach their motor milestone earlier. How the authors address the motor milestone taking the family genetic background in mind?

Response point 4: This is a very interesting issue. However, the aim of our study was to examine changes over the past twenty years in motor milestone achievement and growth in body weight and whether these changes in overweight could be explained by later motor milestone achievement. Although the genetic background is interesting, we did not have the intention of studying the genetics of the relations but specifically in timetrends. In addition, it is assumable that the genetic background did not change over the past 20 years and therefore would not influence the results of our study.

Point 5: In the methods, authors mentioned that data had been collected for majority of twins selected in this study. But what majority means in percentage?

Response point 5: We realize that the word majority is confusing. What was meant was that the majority of children born as twins in the Netherlands were included in the cohort, indicating good coverage of the twin population. However, this is less relevant for the current paper. We removed the words “the majority of” [Line 77].

Point 6: Why authors used twins for the study?

Response point 6: Unfortunately, the twin cohort we used did not include singletons. Of course, data on singletons would add extra information regarding generalizability.  However, despite this lack of singletons, we considered it worthwhile to perform this analysis. The Young Netherlands Twin Register (YNTR) is a cohort study with continuous influx of new-born twins since 1986, resulting in a unique dataset to study changes in development over time, with data from newborns starting in 1987. Although there are cohorts with singletons collecting data on motor milestones and body weight, these studies are mostly closed cohorts, meaning they include children over a period of 1 to 3 years, and then follow them up over time. We added in the introduction the text “because the YNTR is a unique cohort with a continuous influx of new-born twins since 1987” [Line 71].

Point 7: Did the family sign an agreement with the survey?

Response point 7: All data used in the analyses were collected under protocols that have been approved by the appropriate ethics committees, and this study was performed in accordance with the ethical standards from the 1964 Declaration of Helsinki and its later amendments. During registration a consent was signed by the parents/caregivers. For future surveys, the ethics committees accepts surveys sent back voluntarily as consent.

Point 8: The authors didn’t mention whether the study analyzes or compares normal delivery with C-section. Are there any differences?

Response point 8: We thank you very much for this interesting question. We did not studied the motor milestones separate for twins born with or without C-section. In our study population, 30.1% of the twins were born by C-section. With the execution of sitting without support, there were no statistical differences in motor milestones achievement for twins born by C-section or via normal delivery. For sitting without support, twins born by C-section were slightly faster compared to those born via normal delivery (8.6 ± 2.0 and 8.7 ± 1.9 respectively).

Point 9: In the methods, as a suggestion, authors should include a new table organizing the survey during ages;     

Response point 9: We added a table in the methods and therefore also adjusted the numbers of the other tables [Line 86].

Table 1 Data collection of twins from the Young Netherlands Twin Register (YNTR)

Survey

Age 0-6 months

Pregnancy and birth

Table for keeping track on motor milestones

Age 2 years

Growth and motor milestone achievement

Age 4-10 years

Growth and socioeconomic status

Point 10: In the conclusion, line 235, “Ove the same are…” should be “Over the…”.

Response point 10: We added the “r” to “over”.

Round 2

Reviewer 1 Report

There has been a lot of work in this paper, including the proposed suggestions.

There are several issues that if further amended, would improve the quality of the paper.

  1. The study including the title should focus on twins only. The findings of the authors cannot be extrapolated in singletons.
  2. The issue of the “random selection” should be further acknowledged in the limitations section of the study.
  3. Through the comment on the “lack of association of the findings and confounding factors”, I would suggest a further analysis on the special characteristics of each woman (pathologies during pregnancy, way of conceiving, age, BMI, way of delivery (is it really 30%? - bravo), etc) to the final outcome; or express this issue in the limitations section.

Author Response

Response to Reviewer 1 

Dear reviewer, 

Thank you for responding on our revision and for taking the effort to give suggestions to improve our paperWe want to respond to your suggestions in the text below. 

There are several issues that if further amended, would improve the quality of the paper. 

Point 1: The study including the title should focus on twins only. The findings of the authors cannot be extrapolated in singletons. 

Response point 1: Although twins differ from singletons in many perinatal factors, these differences tend to get smaller and smaller with age, especially when you exclude the children who were preterm or small-for-gestational age. Already from the age of two, many differences are made up for. We showed that within the normal window of achievement of motor milestones there are no major differences between twins and singletons (Brouwer et al. 2006). An important aspect is that the differences (in healthy twins) are mostly explained by restriction of intrauterine space and gestational age. Therefore, based on our previous comments, we see, after adjusting for gestational age, the analysis of motor milestones from twins not as a limitation. Therefore we included the text: We expect that the outcomes can be generalized towards singletons because within the normal window of achievement of motor milestones there are no major differences between twins and singletons, as we have shown before [8]. Differences in the age of milestone achievement, are mainly  explained by the shorter GA of twins, therefore when studying motor milestones in our twin population, we adjusted for GA. Furthermore, babies born pre-term (before 32 weeks of GA) or with very low BW (<1500 grams) are considered at risk for motor development deficits [31] and are not included in our studies [line 259-265], in the discussion and not in the limitation 

Based on our own comparison with the Gecko cohort, (please also see our earlier replies) no firm conclusions can be drawn about the differences between twins and singletons in the exact moment of achieving walking without support. The age of “walking without support” was dependent on year of birth and the range of gestational age (for example 36-41 weeks or 36-38 weeks, or comparison with exact matching on gestational age rounded to half a week) and varied from 0.1 to 1.0 months. Singletons from the Gecko Drenthe cohort achieved walking without support with 14.1 ± 1.9 (mean gestational age 39.9 ± 1.6) while twins were able to walk with 14.8 ± 2.3 (mean gestational age 37.0 ± 2.0), varying from 14.5 ± 2.1 to 15.3 ± 2.4 when born in 1987-2000 and 2005-2007 respectively.   

To accommodate the discussion of focusing on twins in our study, we changed our title into: “Infant motor milestones and childhood overweight: trends over two decades in a large twin cohort [line 2-3). 

  1. Brouwer SI, van Beijsterveldt TC, Bartels M, Hudziak JJ, Boomsma DI: Influences on achieving motor milestones: a twin-singleton study. Twin Res Hum Genet 2006, 9(3):424-430.

31: Allen MC, Alexander GR: Gross motor milestones in preterm infants: correction for degree of prematurity. J Pediatr 1990, 116(6):955-959.] 

Point 2: The issue of the “random selection” should be further acknowledged in the limitations section of the study. 

Response point 2: A total of 18.514 infants participated in our study. Observations were independent and the random selection was not based on any child characteristic like birth order, heaviest baby, gender of opposite-sex twin etc It remains unclear whether random selection should be discussed in the limitations section Doing this random selection is a strength rather than a limitation, because it avoids observations being dependent, or clustered. This is usually a problem in twin studies and needs multilevel analyses to adjust for this. As this is a very large cohort, we can perform the random inclusion and still have enough power for the research questions addressed, avoiding uncertainty on whether the clustering is adequately adjusted for. 

Point 3: Through the comment on the “lack of association of the findings and confounding factors”, I would suggest a further analysis on the special characteristics of each woman (pathologies during pregnancy, way of conceiving, age, BMI, way of delivery (is it really 30%? - bravo), etc) to the final outcome; or express this issue in the limitations section. 

Resonse point 3: We confirm that in our study population, 30.1% of the twins were born by C-section, this reflects Dutch policies with respect to delivery. With the exception of sitting without support, there were no statistical differences in motor milestones achievement for twins born by C-section or via normal delivery. For sitting without support, twins born by C-section reached this slightly earlier compared to those born via normal delivery (8.6 ± 2.0 and 8.7 ± 1.9 respectively). We agree with the reviewer that delving into yet more analyses and extending our search by including maternal characteristics might be of interest, but currently believe this to be outside the scope of the present paper. Therefore, we add to the limitations the text: “Furthermore, including maternal characteristics like age, BMI and way of conceiving, might be of interest but are currently outside the scope of the present paper.”[Line 330-331].